# Characterisation of *Pythium aristosporum* Oomycete—A Novel Pathogen Causing Rice Seedling Blight in China

**DOI:** 10.3390/jof8090890

**Published:** 2022-08-23

**Authors:** Jinxin Liu, Ruisi Zhang, Chuzhen Xu, Chunlai Liu, Yanyan Zheng, Xue Zhang, Shasha Liu, Yonggang Li

**Affiliations:** 1College of Agricultural, Northeast Agricultural University, Harbin 150030, China; 2Institute of Plant Protection, Heilongjiang Academy of Agricultural Sciences, Harbin 150030, China

**Keywords:** *Pythium aristosporum*, identification, biological characteristics, host range, fungicide efficacy

## Abstract

Rice seedling blight is a globally occurring seedling disease caused by multiple pathogens. It is currently the most common disease affecting rice production in northeast China; hence, determining the causal agents, including its biological characteristics, host range, and fungicide efficacy is vital for its effective management. The present study obtained 45 pathogenic isolates from diseased rice seedlings in Suihua, Heilongjiang Province, China. Of these, five pathogens were identified based on their morphology and molecular identification, while 10 oomycete isolates were identified as *Pythium aristosporum—*the first to be reported in rice seedling blight. Its optimum growth conditions include a temperature of 25 °C, pH 6, and photoperiod of 24 h. Except for soybean (*Glycine max* (L.) Merr.), black soybean (*Glycine max var.*), and cucumber (*Cucumis sativus* L.), *P. aristosporum* can potentially infect and cause seedling blight on other hosts, such as wheat (*Triticum aestivum* L.), maize (*Zea mays* L.), sorghum (*Sorghum bicolor* (L.) Moench), alfalfa (*Medicago sativa* L.), oats (*Avena sativa* L.), and white clover (*Trifolium repens* L.). Its isolates were found to be highly sensitive to metalaxyl + propamocarb (EC_50_ = 0.0138 μg/mL) with 84.1% efficacy at 313 μg/mL. These results can serve as the basis for controlling *P. aristosporum*.

## 1. Introduction

Rice, wheat, and maize are three of the major crop staples worldwide [1,2]. Among them, rice is the most important cereal crop cultivated in China as it is critical for the country’s agricultural economy [3,4]. Heilongjiang Province is the northernmost rice-growing region and one of the largest commercial Japonica rice production areas [5]. Therefore, ensuring the sustainability of rice production systems in these areas is vital for ensuring global food security and protecting livelihoods [6]. Rice seedling blight, which is a seedling disease caused by diverse pathogenic microorganisms, is the most common disease affecting rice production in northeast China, as low temperatures of 3–7 °C provide favorable conditions for its prevalence [7,8]. Rice seedling blight causes growth inhibition in rice seedlings, which in turn, reduces grain quality and rice yield. Recently, its incidence has increased in northeast China [9], in which stand loss ranges from 10 to 20%, and in severe cases, up to 60–80% [10].

Breeding disease-resistant varieties is considered an effective way to control rice diseases, yet, these have been limited to date [11]. Hence, fungicides have largely remained the most commonly used method to control rice seedling blight [8]. However, these methods vary depending on the complexity and variability of pathogens across different rice-planting areas [12,13]. As rice seedling blight is caused by several pathogenic microorganisms, including *Fusarium* spp. [8,14], *Pythium* spp. [15,16], *Burkholderia* spp. [17,18], *Rhizoctonia solani* [19,20], *Marasmius graminum* [21], and *Curvularia coatesiae* [8], identifying the specific pathogenic species and their occurrence is essential for selecting effective chemical fungicides for appropriate disease prevention and control [7].

Crop rotation is another method to control rice seedling blight, which involves enhancing the disease suppression property of soils against soil-borne plant pathogens [22]. Globally, crops rotated along with rice include alfalfa [23], wheat [24], and maize [25], and the occurrence of pathogens in these hosts has not been systematically analysed for rice seedling blight. Potentially, such analysis can provide a theoretical basis for using this method to reduce rice seedling blight in Heilongjiang Province, China.

In a study of the population structure of pathogens causing rice seedling blight in the same location, a novel pathogenic oomycete was found, having been recovered from diseased seedlings. Hence, this study aims to identify the pathogenic species causing rice seedling blight and analyse its biological characteristics, host range, and sensitivity to fungicides. The results can potentially help farmers and researchers to develop and formulate improved control strategies for preventing and controlling rice seedling blight in the future.

## 2. Materials and Methods

### 2.1. Isolation and Pathogenicity of Pathogens

In May 2019, diseased rice seedlings (cv. Suijing 18) with withered tips, chlorosis, stunting, yellow leaves, leaf drop, crown rot, and inhibited root growth were observed in Suihua City, Heilongjiang Province, China (46.63° N 126.98° E). The soil is mainly black, with a sandy clay texture and an organic matter content of 4–5%. Rice seedlings (*n* = 45) with symptoms of rice seedling blight were randomly collected from three fields with a total area of ~3 ha. The disease incidence ranged from 10–15%, which allowed for pathogen isolation and identification. The symptomatic stem base tissues were surface-disinfected with 0.5% NaOCl for 2 min, rinsed three times in sterile distilled water, cultured on potato dextrose agar (PDA) at 25 °C for three days, then subcultured by transferring hyphal tips onto a V8 agar medium [26]. Thereafter, four diseased stem base tissues were selected from each diseased seedling for pathogen isolation. This was followed by the calculation of the number of pathogens recovered, including the percentage of each genus.

The submerged root technique [27] was used to assess the pathogenicity of the recovered isolates. All isolates were re-isolated from diseased rice plants and observed and assessed based on Koch’s postulates. The roots of 15-day-old rice seedlings (cv. Longdao 18) were submerged for 10 min in a suspension containing 10^6^ zoospores (or conidia)/mL [26] of each isolate. Meanwhile, isolates without zoospores or conidia were purified by selecting a single hyphal tip [28], and a mycelium suspension was prepared using sterile water. The inoculated seedlings were then transplanted into pots containing sterile soil, with ten seedlings each. For each isolate, three pots were used, in which seedlings soaked in sterilised distilled water were used as the control samples. Subsequently after 20 days, symptoms on the inoculated seedlings were observed to determine whether they were consistent with those of the diseased samples in the field. The experiment was repeated thrice under the same conditions.

Disease severity was visually scored based on an assessment of the growth status of the rice seedlings. A scale from 0 to 4 was used [8]: 0 = no symptoms; 1 = small lesions covering less than 1/4 of the stem surface area; 2 = moderate-sized lesions covering approximately 1/4 to 1/2 of the stem surface area; 3 = large lesions covering approximately 1/2 to 3/4 of the stem surface area; and 4 = dead plants with lesions covering the whole stem surface area. The disease index was calculated as Σ (number of diseased rice seedlings at each scale × relative grade) / (total number of surveyed plants × highest disease rate) × 100. The pathogenicity of the strains was then described based on the average disease index of the three experiments, categorised as: weakly pathogenic, disease index < 50; moderately pathogenic, 50 ≤ disease index < 60; and highly pathogenic, disease index ≥ 60.

### 2.2. Identification of the Pathogens Recovered from Diseased Rice Seedlings

Isolates causing rice seedling blight were identified based on their morphological [29] and molecular characteristics. For the latter, the genomic DNA was extracted from the mycelia of representative isolates using a Tiangen Genome Extraction Kit (Tiangen Biotech, Beijing, China). The internal transcribed spacer (ITS) region and cytochrome oxidase subunit II (*CoxII*) genes were amplified using primer pairs ITS1/ITS4 [30,31] and COX2f/COX2r [32], respectively. A polymerase chain reaction (PCR) was performed in a final volume of 50 µL with 10 µM of each primer, 2 × Taq Master Mix, and 10 ng of template DNA. The PCR conditions were as follows: initial denaturation for 5 min at 94 °C, followed by 35 cycles for 1 min at 94 °C and 55 °C, 1.5 min at 72 °C, and final extension for 10 min at 72 °C. The PCR products were purified and sequenced by Shanghai Biological Engineering Co. Ltd. (Shanghai, China). Phylogenetic trees of representative isolates were constructed using PhyML 3.0 (LIRMM, Montpellier, France), based on the maximum likelihood principle [33].

### 2.3. Biological Characteristics of P. aristosporum Isolates

To determine the pH level, temperature, and photoperiod at various mycelial growth rates, the growth rate of each isolate (*n* = 10) was measured on V8 agar at different pH levels (4.0, 6.0, 7.0, 8.0, and 10.0), temperatures (10 °C, 20 °C, 25 °C, 28 °C, 30 °C, and 35 °C), and photoperiods (light 24 h, light/dark = 12 h/12 h, and dark 24 h). A mycelial plug (0.7 cm diameter) of an isolate grown on V8 agar for 96 h was then transferred to a treatment plate with V8 medium and incubated under similar conditions. Each treatment was performed in triplicates, and the entire experiment was repeated twice. The colony diameters of *P. aristosporum* isolates were measured after 72 h.

### 2.4. Host Range Determination of P. aristosporum Isolates

Ten *P. aristosporum* isolates recovered from seedlings with disease symptoms were inoculated on the seedlings of other crops grown in Heilongjiang such as wheat (cv. Longfumai 10), maize (cv. Suiyu 7), sorghum (cv. Suiza 7), alfalfa (cv. Xinjiangdaye), oats (cv. Baiyan 6), white clover (cv. Mini-BL), soybeans (cv. Suinong 26), black soybean (cv. Heizhenzhu), and cucumber (cv. Changchunmici). Seedlings, with ten plants per treatment, were inoculated with a zoospore suspension (1 × 10^6^ zoospores/mL) of the isolate using the soaked root method [34]. Ten seedlings from each crop were soaked in sterile distilled water as control seedlings. Each treatment was replicated thrice. Approximately seven days following inoculation, disease severity was visually scored based on the abovementioned procedure. Isolates of *P. aristosporum* were re-isolated and identified from the inoculated seedlings based on Koch’s postulates. The experiments were conducted in triplicates.

### 2.5. Sensitivity of P. aristosporum Isolates to Fungicides

The mycelial growth rate method [35] was used to assess the sensitivity of *P. aristosporum* against the following fungicides: metalaxyl + hymexazole (30% AS) (Sino-Agri Leading Biosciences Co., Ltd., Tianjin, China), fosetyl-Al (80% WP) (Limin Chemical Co., Ltd., Xinyi, China), and metalaxyl + propamocarb (25% WP) (Jiangsu Baoling Chemical Co., Ltd., Nantong, China).

Each fungicide was added to the V8 agar separately at final concentrations of 0.1 μg/mL, 0.3 μg/mL, 0.5, 1, and 2 μg/mL. A mycelial plug (0.7 cm-diameter) of each isolate (*n* = 10) was placed at the center of a fungicide-amended V8 agar plate and incubated in the dark at 25 °C for 72 h. Sterilised distilled water (1 mL) was added to the V8 medium (500 mL) as a blank control. Each treatment was performed in triplicate and the entire experiment was conducted twice. The colony diameter was measured to evaluate the inhibition of isolate growth. Percent growth inhibition was calculated as 1 − [(diameter of treated colonies − 0.5) / (diameter of control colonies − 0.5) × 100] [36]. The EC_50_ values were estimated using GraphPad Prism 8 (GraphPad Software Inc., San Diego, CA, USA). 

### 2.6. Efficacy of Metalaxyl + Propamocarb against Rice Seedling Blight Caused by P. aristosporum

All experiments were conducted twice under similar conditions. Analysis of variance (ANOVA) was performed using SPSS Statistics 17.0 (IBM/SPSS, Armonk, NY, USA), and treatment means were separated using Duncan’s multiple range test (*p* = 0.05).

Pot experiments were performed in 20 cm-diameter plastic pots situated in a greenhouse under a temperature of 25 ± 3 °C, and a photoperiod of 12 h/12 h (light/dark). The greenhouse was located at the experimental station of the Northeast Agricultural University, Harbin, China. Rice seedlings were treated with 10 mL of *P. aristosporum* zoospore suspension (10^6^ zoospores/mL) following the germination of rice seeds in each pot. The treatments used were applied to the soil at the two-leaf stage, which were repeated after ten days. Metalaxyl + propamocarb (200 mL) at concentrations of 313, 250, 208, and 0 μg/mL was uniformly applied to the soil surface of pots containing rice seedlings. The seedlings were watered uniformly daily using overhead irrigation to maintain soil moisture. Disease severity was visually scored 15 days following the second fungicide treatment, as described above. The experiments were repeated twice. Seedling height and fresh weight were measured [37]. The disease index and control efficacy were calculated using the following formulas:

Disease index = Σ (number of diseased rice seedlings at each scale × relative grade) / (total number of surveyed plants × highest disease rate) × 100. 

% Control efficacy = [(disease index of control group − disease index of fungicide treatment group) / disease index of control group] × 100.

### 2.7. Data Analysis

All experiments were conducted twice under the same conditions. ANOVA was performed using SPSS Statistics 17.0 (IBM/SPSS, Armonk, NY, USA). The treatment means were separated using the least significant difference test (*p* ≤ 0.05).

## 3. Results

### 3.1. Identification of Causal Organisms

A total of 45 isolates were obtained from 45 symptomatic seedlings, and their pathogenicity was verified according to Koch’s postulates. Their morphological and molecular identification indicated that they belonged to the following species: *Fusarium oxysporum* (51.1% of isolates), *P. aristosporum* (22.2% of isolates), *Fusarium redolens* (13.3% of isolates), *Fusarium solani* (6.7% of isolates), *and Rhizoctonia solani* (6.7% of isolates) (Table 1).

### 3.2. Pathogenicity of P. aristosporum on Rice

Differences in pathogenicity were detected among the ten isolates of *P. aristosporum*, however, all were found to be pathogenic to rice. Categorically, four isolates (JX8, JX18, JX22, and SH1) were highly pathogenic, whereas the remaining were moderately pathogenic to the evaluated rice cultivar (Table 2).

### 3.3. Identification of P. aristosporum

The ten *Pythium* isolates were observed to have aseptate hyphae with white cottony growth on V8 agar plates (Figure 1A). The sporangia were finger- or lobe-like, which formed germ tubes and zoospores at room temperature and 10–15 °C, respectively. The oogonium was subglobose with an average diameter of 27.2 μm, and ranging from 19–35.5 μm. The antheridia were clavate or had curved necks, which were approximately 15 μm × 6 μm, and were in contact with the oogonium at the top. Each oogonium had 3–10 antheridia and 1–2 oospores (Figure 1C–E). The oospores were spherical, smooth, and aplerotic, with an average diameter of 19.9 μm, and ranging from 13.5 to 26.3 μm. These were also either unfilled or filled with organelles. All ten isolates were identified as *Pythium* spp. based on their cultural and micromorphological characteristics [37].

As all ten isolates shared similar morphological characteristics. Their genomic DNA was extracted, in which the internal transcribed spacer regions (ITS) and cytochrome oxidase subunit II (*CoxII*) genes were amplified using the primer pairs of ITS1/ITS4 and COX2f/COX2r, respectively. The ITS and *CoxII* sequences were deposited in the GenBank (accession numbers in Table A1). BLAST analysis showed that the sequences obtained for the ITS and *CoxII* amplicons were highly similar to those of *P. aristosporum* and *P. arrhenomanes*. Of the ten isolates, two (JS22 and SH1) were randomly selected to undergo phylogenetic tree construction based on their ITS region genes. This showed that isolates JS22 and SH1 belonged to a similar evolutionary branch as *P. aristosporum* and *P. arrhenomanes*, with a similarity of up to 95% (Figure 2). As *P. aristosporum* has aplerotic oospores, with fewer antheridia per oogonium than *P. arrhenomanes,* the ten isolates were identified as *P. aristosporum* [38].

### 3.4. Biological Characteristics of P. aristosporum

All ten *P. aristosporum* isolates developed normally within the pH range of 4.0–10.0. However, significant differences in mycelial growth at different pH values (*p* < 0.05) were observed. Meanwhile, the optimum pH value was observed at pH 6.0 (Figure 3A). All isolates also developed normally under a temperature range of 10–35 °C. Similarly, significant differences in mycelial growth at different temperatures (*p* < 0.05) were observed. The optimum temperature was observed to be 25 °C (Figure 3B), while the optimum photoperiod was 24 h of light (Figure 3C).

### 3.5. Host Range Determination of P. aristosporum

The *P. aristosporum* isolates were categorised as follows: moderately pathogenic to wheat seedlings, weakly pathogenic to maize and sorghum seedlings, and highly pathogenic to alfalfa, oats, and white clover seedlings (Table 3). Withered tips, chlorosis, stunting, dried leaves, and crown rot were the common symptoms observed on wheat seedlings; withered tips, chlorosis, stunting, and yellow leaves were observed on maize and sorghum seedlings. Withered tips, chlorosis, stunting, yellow leaves, leaf drop, crown rot, and even death were observed in alfalfa, oats, and white clover seedlings (Figure 4A–I). Meanwhile, no symptoms were observed in soybean, black soybean, and cucumber seedlings. All *P. aristosporum* isolates inoculated on the other crop seedlings were successfully re-isolated from the inoculated wheat, maize, sorghum, alfalfa, oats, and white clover seedlings, whereas they could not be isolated from soybean, black soybean, or cucumber seedlings.

### 3.6. Efficacy of Chemical Fungicides

All ten *P. aristosporum* isolates showed consistent sensitivity to metalaxyl + hymexazol, fosetyl-Al, and metalaxyl + propamocarb. Metalaxyl + propamocarb had the strongest inhibitory effect on *P. aristosporum* growth in vitro, whereas metalaxyl + hymexazol had the weakest (*p* < 0.05). The EC_50_ value of metalaxyl + propamocarb was the lowest at 0.0138 μg/mL, followed by that of fosetyl-Al, with an EC_50_ value of 0.5647 μg/mL. The EC_50_ value of metalaxyl + hymexazole was the highest (0.5952 μg/mL) (Figure 5).

### 3.7. Efficacy of Metalaxyl + Propamocarb on Rice Seedling Blight Caused by P. aristosporum

Metalaxyl + propamocarb at 313, 250, and 208 μg/mL exhibited excellent control of seedling blight (*p* < 0.05), with control efficacies of 84.1%, 80.4%, and 75.9%, respectively. In addition, the average plant height, root length, and fresh weight were significantly greater in all treated plants than in the control (*p* < 0.05) (Table 4).

## 4. Discussion

Rice seedling blight is responsible for the severe decrease in rice yield and quality in many countries [39,40], and various pathogens, including *F. oxysporum*, *F. solani*, *R. solani* [41], and *F. redolens* are associated with this disease, having already been previously isolated from diseased rice seedlings [27]. However, this is the first reported case where *P. aristosporum* has been described as a direct causal agent of rice seedling blight. As rice is the main cereal crop cultivated in this region, the occurrence of the disease is a serious threat to rice production, hence, prevention and control are necessary to sustain the economic contribution of rice production. Thus, characterising *P. aristosporum* is crucial for understanding the causes of the disease, including its occurrence and epidemics, and for formulating more scientific and appropriate prevention strategies.

In this study, morphological characteristics and molecular identification were simultaneously used to ensure the accuracy and reliability of the results, as only the former has been used as the sole identifier of *Pythium* species in the past [38,42]. In addition, molecular identification is a useful reinforcement when morphological characteristics are overlapping and species determination becomes excessively time-consuming [43,44]. With the growing deposition of sequence data in publicly accessible databases, the ITS region of the nuclear DNA has become one of the most widely used loci to identify *Pythium* spp. [45,46], in which ITS sequences have been specifically used to identify the *Pythium* genus and species associated with soybean seedlings [47], wheat [31], and *P. arrhenomanes* isolated from rice in China [16]. However, the molecular identification of *P. arrhenomanes* and *P. aristosporum* is inconclusive with ITS, considering that morphological traits are key to distinguishing the two species [16,46]. Therefore, identifying these characteristics is crucial for detecting *P. aristosporum*. Results suggest that the isolates were identified to be *P. aristosporum* as aplerotic oospores and fewer antheridia per oogonium than *P. arrhenomanes* were recovered from the isolates [38].

The rice seedling blight disease commonly occurs during the seedling stage, particularly when seedlings are grown in greenhouses [48]. Based on the observed biological characteristics of *P. aristosporum* causing rice seedling blight, its optimum growth conditions were in temperatures ranging from 25–30 °C, pH levels from 6.0 to 7.0, and a photoperiod of 24 h. These conditions were consistent with those previously used to produce rice seedlings in northeastern China [49,50], in which it has been previously demonstrated that Pythium activity is easily affected by environmental factors such as light, temperature, and pH [51,52,53].

*Pythium aristosporum* is pathogenic to wheat, maize, sorghum, alfalfa, oats, and white clover seedlings. Conversely, it is not pathogenic to soybean, black soybean, or cucumber seedlings. *Pythium* spp. are known for having a broad host range and causing pre- and post-emergence seedling damping-off, as well as root, seed, or fruit rot in almost all cereals, including wheat, maize, and rice [38,54,55]. Similarly, a wide geographic distribution, host range, and severe yield loss have been reported for *P. aristosporum* [56]. For example, the pathogen causes root diseases of wheat, oats, barley, and rye in Canada [57], snow rot of winter wheat (*T. aestivum* L.) in Washington [58], root rot of snap beans (*Phaseolus vulgaris* L.) in Wisconsin [59], root rot of konnyaku (*Amorphophallus konjac* C. Koch) in Ibaraki [60], root dysfunction of creeping bentgrass in Maryland [28], blight on turf grass in Italy [61], and stalk rot in maize (*Z. mays*) in China [62]. The wide host range of *P. aristosporum* can be potentially related to the crop production in northeast China, as the results from this study suggest that rotating rice with soybean, black soybean, or cucumber helps avoid large-scale outbreaks of diseases caused by *P. aristosporum* in rice seedling fields.

The management of this disease mainly involves the use of fungicides [28]. In this study, three chemical fungicides, namely: metalaxyl + hymexazol, fosetyl-Al, and metalaxyl + propamocarb, which are all widely used in rice production in northeastern China, were selected. Results demonstrated that metalaxyl + propamocarb had the strongest inhibitory effect on the growth of *P. aristosporum,* followed by metalaxyl + hymexazol, based on their sensitivities against the selected bacteria. In the pot experiment in the greenhouse, results showed that 313 μg/mL metalaxyl + propamocarb reduced disease occurrence by 84.1%. It also enhanced the quality of rice seedlings. Metalaxyl was previously reported to inhibit the development of *Pythium* spp., control wheat root rot caused by *P. aristosporum*, and promote wheat growth [63]. Propamocarb was also used to protect turfgrass against *Pythium* spp. in Italy [64]; however, other *Pythium* strains with low sensitivities to propamocarb have been reported in several golf courses [28]. In the present study, *P. aristosporum* isolates recovered from diseased rice seedlings demonstrated high sensitivities to metalaxyl and propamocarb, owing to the occurrence of *P. aristosporum* as a novel pathogen causing rice seedling blight, in which the isolates did not develop resistance to metalaxyl and hymexazol. Mixing the fungicides has also been shown to improve the efficacy and delay the development of resistance in pathogens to a certain extent [11]. Therefore, metalaxyl + propamocarb can potentially be used to control rice seedling blight caused by *P. aristosporum*. However, it is recommended that further research be conducted to further identify the appropriate method and period of application of metalaxyl + propamocarb.

## 5. Conclusions

This is the first report that describes *P. aristosporum* as the main cause for the occurrence of rice seedling blight in northeastern China. It is inferred that the environment within the region is favorable for the development and prevalence of *P. aristosporum*, in which it poses a potential risk to regional rice production and global food security. Thus, investigating the occurrence of the novel disease caused by *P. aristosporum* must be considered in the development of improved disease management strategies to ensure healthy growth and development of rice seedlings.

## Figures and Tables

**Figure 1 jof-08-00890-f001:**
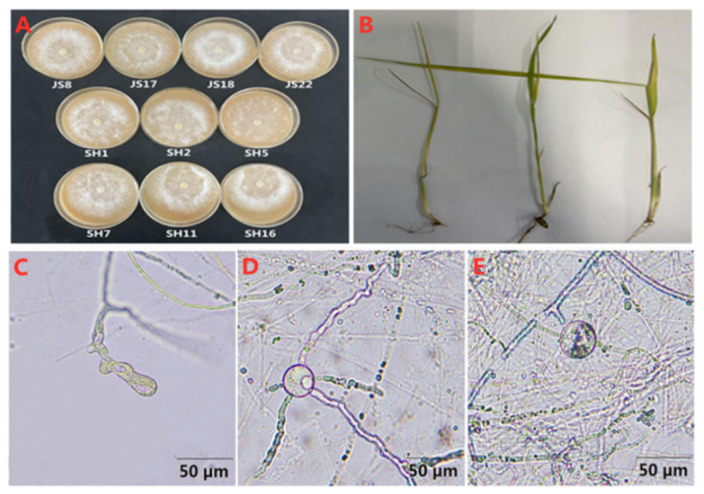
*Pythium aristosporum* causing rice seedling blight. (**A**) colony morphology of *P. aristosporum* isolates on V8 agar plates, (**B**) rice seedling blight caused by *P. aristosporum*, (**C**) sporangium, (**D**) oogonium, antheridium, (**E**) Oospore.

**Figure 2 jof-08-00890-f002:**
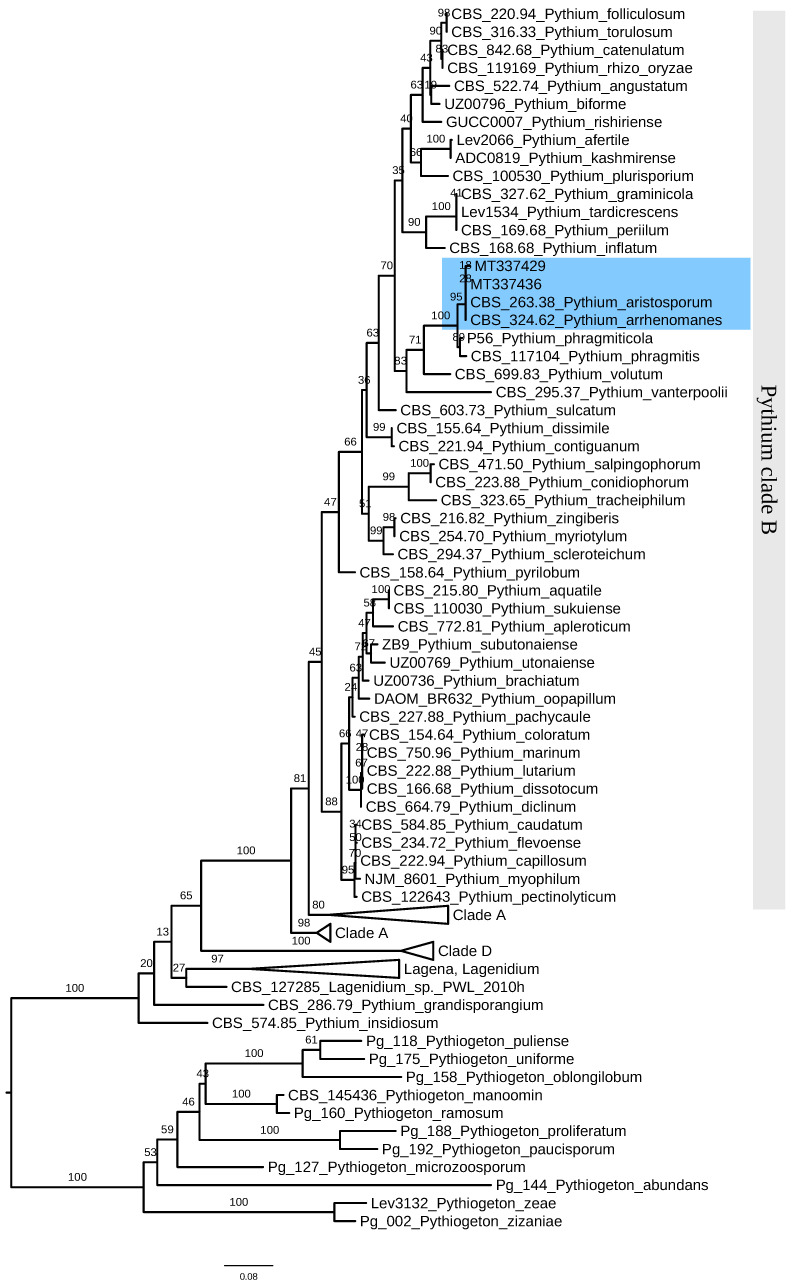
Phylogenetic tree of *Pythium aristosporum* strains JS22 and SH1 of rice seedling blight based on internal transcribed spacer regions (ITS) gene. The bootstrap values on the branching nodes were calculated on 1000 replications. The scale bar indicated 0.08 substitutions per nucleotide position.

**Figure 3 jof-08-00890-f003:**
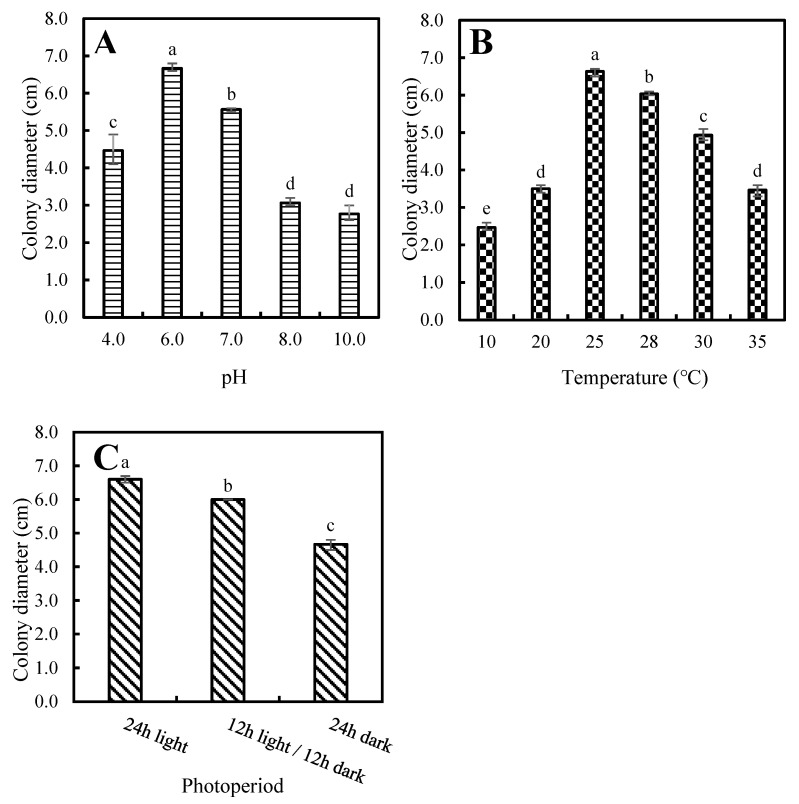
Colony diameters of ten *Pythium aristosporum* isolates at different treatments. (**A**) pH values. (**B**) temperatures. (**C**) photoperiods. The letters above the bars indicate significant difference for each isolate according to the least significant difference test (*p* = 0.05).

**Figure 4 jof-08-00890-f004:**
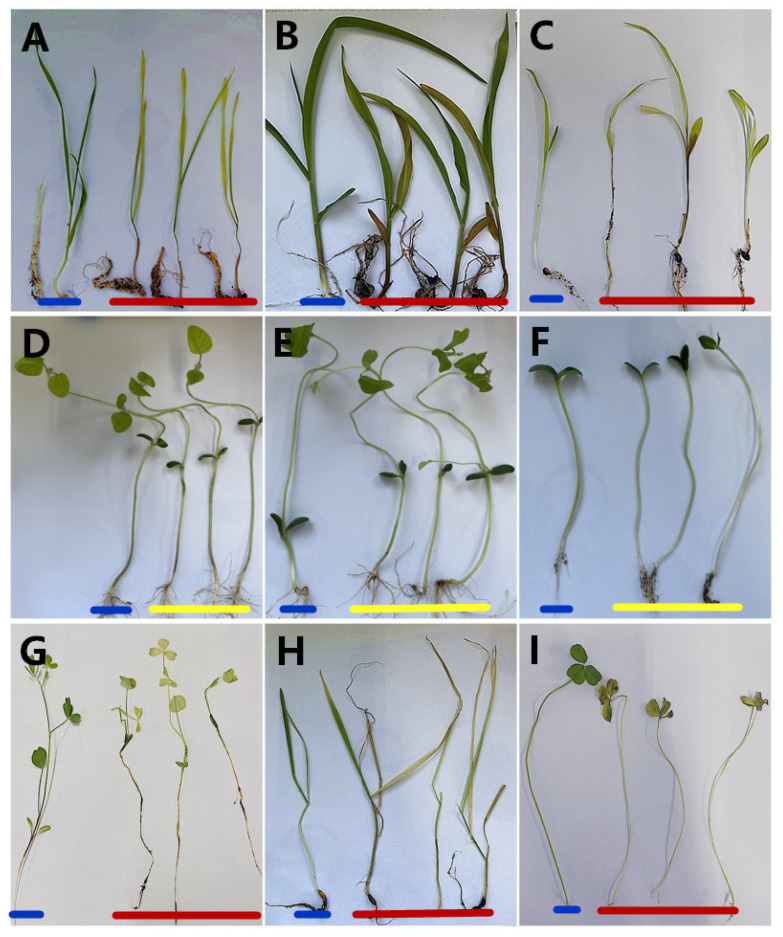
Pictures of various crop plant seedlings acting as either hosts or non-hosts of *Pythium aristosporum*. (**A**) wheat (*Triticum aestivum* L.), (**B**) maize (*Zea mays* L.), (**C**) sorghum (*Sorghum bicolor* (L.) Moench), (**D**) soybean (*Glycine max* (L.) Merr.), (**E**) black soybean (*Glycine max var.*), (**F**) cucumber (*Cucumis sativus* L.), (**G**) alfalfa (*Medicago sativa* L.), (**H**) oats (*Avena sativa* L.), (**I**) white clover (*Trifolium repens* L.), In each picture, the blue line indicates the control plant; the red line indicates the diseased plants inoculated with *P. aristosporum*; the yellow line indicates the uninfected plants inoculated with *P. aristosporum*.

**Figure 5 jof-08-00890-f005:**
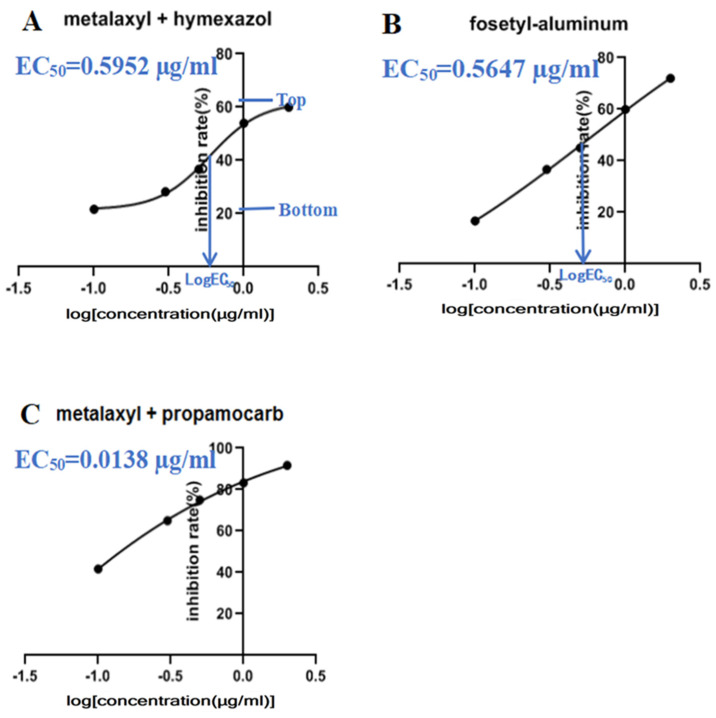
Three-parameter log-logistic estimated EC_50_ values for *Pythium aristosporum* isolate, challenged against common chemical fungicides used to manage rice seedling blight in Northeast China. (**A**) metalaxyl + hymexazol, (**B**) fosetyl – aluminum, (**C**) metalaxyl + propamocarb.

**Table 1 jof-08-00890-t001:** Identification of seedling blight pathogens (*n* = 45) infecting rice in Heilongjiang province, China.

**Pathogens**	**No of Isolates**	**Frequency (%)**
*Fusarium oxysporum*	23	51.1
*Pythium aristosporum*	10	22.2
*F. redolens*	6	13.3
*F. solani*	3	6.7
*Rhizoctonia solani*	3	6.7

**Table 2 jof-08-00890-t002:** Pathogenicity of isolates of *Pythium aristosporum* isolated from rice seedlings in Heilongjiang province, China.

No.	Isolate	Disease Index	Pathogenicity ^a^
1	JS8	80.00	H
2	JS17	58.33	M
3	JS18	80.00	H
4	JS22	86.67	H
5	SH1	85.00	H
6	SH2	56.67	M
7	SH5	51.67	M
8	SH7	58.33	M
9	SH11	58.33	M
10	SH16	58.33	M

^a^ H = highly pathogenic, M = moderately pathogenic.

**Table 3 jof-08-00890-t003:** Pathogenicity of isolates of *Pythium aristosporum* obtained from rice seedlings in Heilongjiang province, China.

Isolates	Wheat	Maize	Sorghum	Alfalfa	Oats	White Clover
JS8 ^a^	56.67 (M)	41.67 (W)	21.67 (W)	95.00 (H)	88.33 (H)	78.33(H)
JS17 ^a^	51.67 (M)	36.67 (W)	16.67 (W)	95.00 (H)	91.67 (H)	71.67 (H)
JS18 ^a^	55.00 (M)	45.00 (W)	21.67 (W)	95.00 (H)	88.33 (H)	75.00 (H)
JS22 ^a^	58.33 (M)	48.33 (W)	25.00 (W)	98.33 (H)	90.00 (H)	75.00 (H)
SH1 ^a^	58.33 (M)	46.67 (W)	23.33 (W)	96.67 (H)	91.67 (H)	71.67 (H)
SH2 ^a^	51.67 (M)	36.67 (W)	16.67 (W)	95.00 (H)	83.33 (H)	66.67 (H)
SH5 ^a^	51.67 (M)	36.67 (W)	16.67 (W)	91.67 (H)	85.00 (H)	66.67 (H)
SH7 ^a^	51.67 (M)	36.67 (W)	16.67 (W)	95.00 (H)	86.67 (H)	66.67 (H)
SH11 ^a^	51.67 (M)	36.67 (W)	16.67 (W)	95.00 (H)	88.33 (H)	73.33 (H)
SH16 ^a^	51.67 (M)	36.67 (W)	16.67 (W)	95.00 (H)	86.67 (H)	66.67 (H)

^a^ Values in the column indicate the mean disease index of rice seedling blight caused by ten *Pythium aristosporum* isolates. Capital letters in brackets in the column indicate pathogenicity to different crop seedlings (soybeans, black soybeans, and cucumbers were also inoculated, but no symptoms were recorded).

**Table 4 jof-08-00890-t004:** Control effect of metalaxyl + propamocarb on rice seedling blight through pot experiment in a greenhouse.

Fungicide	Effective Dose (μg/mL)	Control Efficacy (%) ^a^	Plant Height (cm) ^a^	Root Length (cm) ^a^	Fresh Weight (g) ^a^
Metalaxyl + propamocarb (25% WP)	313	84.1 ± 0.05 ^a^	12.7 ± 0.05 ^a^	4.6 ± 0.03 ^a^	0.235 ± 0.001 ^a^
	250	80.4 ± 0.29 ^b^	12.1 ± 0.06 ^b^	4.5 ± 0.04 ^a^	0.230 ± 0.000 ^a^
	208	75.9 ± 0.05 ^c^	10.5 ± 0.03 ^c^	4.2 ± 0.03 ^b^	0.221 ± 0.001 ^b^
Control ^b^	-	-	8.9 ± 0.04 ^d^	3.7 ± 0.07 ^c^	0.176 ± 0.004 ^c^

^a^ Values in the column indicate mean ± standard error (SE) of two repeated experiments; values followed by different letters are significantly different according to the least significant difference test (*p* ≤ 0.05). Control ^b^ = not treated with fungicide.

## Data Availability

The data that support the findings of this study are shown in Table A1.

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
