# Peer review of "Characterisation of *Pythium aristosporum* Oomycete—A Novel Pathogen Causing Rice Seedling Blight in China"

_jof, 2022, doi:10.3390/jof8090890_

Round 1
Reviewer 1 Report
In this submission, the authors describe the isolation of Pythium aristosporum as the causal agent of rice seedling blight. Among 45 fungal strains isolated from the stem of rice seedlings showing blight symptoms from a single geographic region, ten of them were identified as Pythium aristosporum by culture and microscopic characteristics, as well as molecular identification using ITS rRNA sequence (for two isolates). Biological characterization such as virulence, host range and sensitivity to fungicides was carried out.
Some concerns and specific comments on the submission:
A more rigorous approach should be followed to unequivocally demonstrate Pythium aristosporum as a novel causal agent of rice seedling blight. Among the ten Pythium isolates, only two were selected for DNA isolation and analysis of the ITS rRNA sequencing. At least, all ten strains in the study should be analyzed by molecular methods, and a larger sample of Pythium isolates from a more diverse geographic region should be included, as well. Also, two or more gene loci should be included for a proper molecular identification. For Peronosporomycetes (Oomycetes), the COX2 gene sequence appears to provide additional information for Pythium phylogenetics (See: Deborah S. S. Hudspeth, Steven A. Nadler & Michael E. S. Hudspeth (2000) A COX2 molecular phylogeny of the Peronosporomycetes, Mycologia, 92:4, 674-684, DOI:10.1080/00275514.2000.12061208).
In Materials & Methods section, the authors cite “A published method [26] was used to isolate the pathogen from symptomatic stem base tissues and culture them on potato dextrose agar…”. However, reference [26] is a short Plant Disease note on Fusarium tricinctum isolate as the causal agent of rice seedling blight, so important methodological details are missing. For instance, did the authors perform mono-zoospore isolation?? and how was this carried out? Such details are of paramount importance to demonstrate the presence of a novel pathogen as a causal agent of plant disease.
Table 3. Eliminate the columns for soybean, black soybean and cucumber. Indicate in the legend that these hosts were inoculated as well but no symptoms were recorded.
English language style, grammar and syntaxis should be thoroughly revised.
Methods details should be corrected: For example, LINE 78 (page 2): “…were submerged for 10 min in a suspension containing 106 zoospores…”. But probably, the authors mean 10^6 spores (i.e. 1,000,000 spores).
LINE 85 (page 2): The experiment conducted three times. But, maybe they mean: The experiment WAS conducted three times. Several errors like this are evident through the manuscript.
Author Response
Response to Reviewer 1 Comments
Point 1: A more rigorous approach should be followed to unequivocally demonstrate Pythium aristosporum as a novel causal agent of rice seedling blight. Among the ten Pythium isolates, only two were selected for DNA isolation and analysis of the ITS rRNA sequencing. At least, all ten strains in the study should be analyzed by molecular methods, and a larger sample of Pythium isolates from a more diverse geographic region should be included, as well. Also, two or more gene loci should be included for a proper molecular identification. For Peronosporomycetes (Oomycetes), the COX2 gene sequence appears to provide additional information for Pythium phylogenetics (See: Deborah S. S. Hudspeth, Steven A. Nadler & Michael E. S. Hudspeth (2000) A COX2 molecular phylogeny of the Peronosporomycetes, Mycologia, 92:4, 674-684, DOI:10.1080/00275514.2000.12061208).
Response 1: All ten isolates were extracted genomic DNA, and the internal transcribed spacer (ITS) region and cytochrome oxidase subunit II (CoxII) gene were amplified and sequenced using the primers ITS1/ITS4 and COX2f/COX2r, respectively. ITS and CoxII sequences of isolates were deposited in GenBank (accession numbers in Table A1). BLAST analysis showed that the obtained sequences for the ITS and CoxII amplicon of the ten isolates high similarity with P. aristosporum and P. arrhenomanes. Therefore, the molecular identification of P. arrhenomanes and P. aristosporum is not conclusive with ITS and CoxII, morphological traits are the key to distinguishing the two species (Robideau et al. 2011; Ling et al. 2018). In addition, this area is a typical representative area of our second cumulative temperature zone in Heilongjiang Province, and is also the main cultivation area of rice, which is geographically different from other areas, so our study is somewhat representative.
It is revised in the revised manuscript. (Line 75-78, 233-244)
Point 2: In Materials & Methods section, the authors cite“A published method [26] was used to isolate the pathogen from symptomatic stem base tissues and culture them on potato dextrose agar…”. However, reference [26] is a short Plant Disease note on Fusarium tricinctum isolate as the causal agent of rice seedling blight, so important methodological details are missing. For instance, did the authors perform mono-zoospore isolation?? and how was this carried out? Such details are of paramount importance to demonstrate the presence of a novel pathogen as a causal agent of plant disease.
Response 2: It is added in the revised manuscript. (Line 107-109, 474-475)
Point 3: Table 3. Eliminate the columns for soybean, black soybean and cucumber. Indicate in the legend that these hosts were inoculated as well but no symptoms were recorded.
Response 3: It is revised in the revised manuscript. (Table 3)
Point 4: English language style, grammar and syntaxis should be thoroughly revised.
Response 4: It is revised in the revised manuscript. (Full text)
Point 5: Methods details should be corrected: For example, LINE 78 (page 2): “…were submerged for 10 min in a suspension containing 106 zoospores…”. But probably, the authors mean 10^6 spores (i.e. 1,000,000 spores).
Response 5: It is revised in the revised manuscript. (Line 84)
Point 6: LINE 85 (page 2): The experiment conducted three times. But, maybe they mean: The experiment WAS conducted three times. Several errors like this are evident through the manuscript..
Response 6: It is revised in the revised manuscript. (Line 91; Full text)
Reviewer 2 Report
The paper is well written and the research conducted in a proper manner. It was a pleasure to read and review the manuscript. I have some comments and advises that could be helpful for the improvement of the presentation.
1. Row 78, p. 2 - have to be 106 instead of 106 zoospores
2. As the morphology is very important for species identification of P. aristosporum it is necessary to precise a description of the microscopic observations. A type of the microscope used and magnification of photos have to be added in MM and/or on Fig.1.
3. Fig.1 A) and B) colony of P. aristosporum isolate JS22 on PDA. Why do you have two photos of one isolate on the same medium? It is better to choose two isolates, for instance picture of the isolate SH1, in addition to JS22.
4. In Conclusions - change to Italic the P. aristosporum everywhere
5. Row 346, p. 11 - Capitals on „northeast china”
Author Response
Response to Reviewer 2 Comments
Point 1: Row 78, p. 2 - have to be 106 instead of 106 zoospores.
Response 1: It is revised in the revised manuscript. (Line 84)
Point 2: As the morphology is very important for species identification of P. aristosporum it is necessary to precise a description of the microscopic observations. A type of the microscope used and magnification of photos have to be added in MM and/or on Fig.1.
Response 2: It is revised in the revised manuscript. (Figure 1C, D, E)
Point 3: Fig.1 A) and B) colony of P. aristosporum isolate JS22 on PDA. Why do you have two photos of one isolate on the same medium? It is better to choose two isolates, for instance picture of the isolate SH1, in addition to JS22.
Response 3: The pictures of ten isolates is added in the revised manuscript.(Figure 1A)
Point 4: In Conclusions - change to Italic the P. aristosporum everywhere.
Response 4: It is revised in the revised manuscript. (Line 392-496)
Point 5: Row 346, p. 11 - Capitals on „northeast china”.
Response 5: It is revised in the revised manuscript. (Line 353)
Round 2
Reviewer 1 Report
The original recommendation was for Extensive editing of English language and style required. However, the revised version only shows moderate editing that does not contribute to improve the text.
Upon my question on the details of the isolation method, the authors replaced the originally cited reference (Li, Y.G.; Zhang, X.; Zhang, R.; Liu, J.X.; Ali, E.; Ji, P.S.; Pan, H.Y. Occurrence of seedling blight caused by Fusarium tricinctum on rice in China. Plant Dis 2019, 10, 1789-1790) with the one by Grijalba et al. (2017) [Ref. 26], resulting in some changes in culture media employed (V8) instead of PDA, which is odd and must be clarified.
Because they are reporting Pythium aristosporum as a novel pathogen in rice seedlings in China, a very rigorous methodological approach must be followed in the isolation and identification.
Author Response
Response to Reviewer 1 Comments
Point 1: The original recommendation was for Extensive editing of English language and style required. However, the revised version only shows moderate editing that does not contribute to improve the text.
Response 1: The manuscript was further polished and revised according to the suggestions of the editors and reviewers. (Attached document certifies)
Point 2: Upon my question on the details of the isolation method, the authors replaced the originally cited reference (Li, Y.G.; Zhang, X.; Zhang, R.; Liu, J.X.; Ali, E.; Ji, P.S.; Pan, H.Y. Occurrence of seedling blight caused by Fusarium tricinctum on rice in China. Plant Dis 2019, 10, 1789-1790) with the one by Grijalba et al. (2017) [Ref. 26], resulting in some changes in culture media employed (V8) instead of PDA, which is odd and must be clarified.
Response 2: The pathogens of rice seedlings with symptoms of rice seedling blight could not be determined before isolation and culture. Therefore, we selected the commonly used tissue isolation method used PDA medium. After preliminary morphological identification, we found that it was Pythium spp. And the isolates were cultured on V8 medium for fulfilling Koch’s postulates and pathogenicity determination, etc. In this paper, we used the same method as Grijalba et al. (2017), i.e., using tissue isolation method to isolate Pythium graminicola, causal agent of kikuyu yellows, cultured on PDA, and subcultured on V8 agar medium.

Round 3
Reviewer 1 Report
The manuscript is improved.
Author Response
Response to Reviewer 1 Comments
Point 1: The manuscript is improved.
Response 1: We have contacted a certified professional company that did a final check of English language and style of the manuscript. (Attached document certifies)
